# A Novel Peptide Ameliorates LPS-Induced Intestinal Inflammation and Mucosal Barrier Damage via Its Antioxidant and Antiendotoxin Effects

**DOI:** 10.3390/ijms20163974

**Published:** 2019-08-15

**Authors:** Lulu Zhang, Xubiao Wei, Rijun Zhang, Dayong Si, James N. Petitte, Baseer Ahmad, Manyi Zhang

**Affiliations:** 1Laboratory of Feed Biotechnology, State Key Laboratory of Animal Nutrition, College of Animal Science and Technology, China Agricultural University, Beijing 100193, China; 2Prestage Department of Poultry Science, North Carolina State University, Raleigh, NC 27695, USA

**Keywords:** anti-inflammatory activity, anti-oxidation, lipopolysaccharide neutralization, Intestinal barrier, NF-κB

## Abstract

Intestinal inflammation is an inflammatory disease resulting from immune dysregulation in the gut. It can increase the risk of enteric cancer, which is a common malignancy globally. As a new class of anti-inflammatory agents, native peptides have potential for use in the treatment of several intestinal inflammation conditions; however, their potential cytotoxicity and poor anti-inflammatory activity and stability have prevented their development. Hybridization has been proposed to overcome this problem. Thus, in this study, we designed a hybrid peptide (LL-37-TP5, LTP) by combing the active centre of LL-37 (13–36) with TP5. The half-life and cytotoxicity were tested in vitro, and the hybrid peptide showed a longer half-life and lower cytotoxicity than its parental peptides. We also detected the anti-inflammatory effects and mechanisms of LTP on Lipopolysaccharide (LPS)-induced intestinal inflammation in murine model. The results showed that LTP effectively prevented LPS-induced weight loss, impairment of intestinal tissues, leukocyte infiltration, and histological evidence of inflammation. Additionally, LTP decreased the levels of tumour necrosis factor-alpha, interferon-gamma, and interleukin-6; increased the expression of zonula occludens-1 and occludin; and reduced permeability in the jejunum of LPS-treated mice. Notably, LTP appeared to be more potent than the parental peptides LL-37 and TP5. The anti-inflammatory effects of LTP may be associated with the neutralization of LPS, inhibition of oxidative stress, and inhibition of the NF-κB signalling pathway. The findings of this study suggest that LTP might be an effective therapeutic agent for treating intestinal inflammation.

## 1. Introduction

Intestinal inflammation is a defensive response to invasion of the host by microbiological toxins (e.g., lipopolysaccharide; LPS) or pathogens (e.g., *Escherichia coli*) [1,2]. Abdominal pain, diarrhoea, rectal bleeding, weight loss, malnutrition, and fever are the common symptoms of intestinal inflammation [3]. Furthermore, intestinal inflammation is an important driver of a wide range of highly fatal diseases such as colorectal and small intestinal cancers [4,5,6]. According to recent studies, dysregulation of the immune response and breakdown of the mucosal barrier are the main mechanisms by which intestinal inflammation occurs [7]. In addition, microbiological toxins are responsible for the synthesis of reactive oxygen species (ROS), and when overproduced may lead to a considerable oxidative stress [8,9]. Moreover, oxidative stress has been proposed to be both a putative causal and perpetuating factor for intestinal inflammation [10].

Corticosteroids are considered the major therapeutic option to treat patients with intestinal inflammation [11,12,13]. These treatments can successfully decrease the production of pro-inflammatory cytokines, cell adhesion molecules, and other key mediators of inflammation [14]; however, prolonged exposure to corticosteroids results in various side effects, such as central adiposity, insulin resistance, and gastrointestinal complications [15,16]. Therefore, it is necessary to develop new drugs that have both the desired efficacy and better safety.

In recent years, LL-37, a human cationic antimicrobial peptide, has been reported to have strong anti-inflammatory effects [17,18,19,20]. LL-37 can not only directly interact with lipopolysaccharide (LPS) to inhibit the inflammatory response [21], but also decrease the activities of intracellular signalling pathways, such as NF-κB pathway, to dampen the inflammatory response [18]. Moreover, LL-37 can upregulate tight junction-related proteins and increases epidermal keratinocyte barrier function, suggesting that it may also ameliorate intestinal inflammation [17]. Thus, LL-37 can potentially prevent or attenuate intestinal inflammation.

Thymopentin (TP5), the Arg32–Tyr36 fragment derived from thymopoietin, is an immunomodulator [22,23]. TP5 exerts its anti-inflammatory effect by inhibiting the transcription factor NF-κB and p38 signalling cascades, which play crucial roles in inflammation [24,25,26]. Additionally, TP5 plays an important role in T-lymphocyte maturation and differentiation and thus can regulate immunity and the inflammatory response [27,28]. Overall, TP5 exhibits immunoregulatory activity and low cytotoxicity. Therefore, it is commonly used in the clinic to treat various types of inflammatory diseases, such as infectious diseases.

LL-37 plays a critical role in the process of anti-inflammation, but its significant cytotoxicity towards eukaryotic cells has hampered its clinical development [29]. In contrast, TP5 shows low cytotoxicity, but it exhibits relatively weak anti-inflammatory activity and a short half-life [30,31]. Hybridization, a simple and effective strategy that combines the advantages of different native peptides [32,33], has been proposed as a method to improve the anti-inflammatory activity, prolong the half-life, and reduce undesirable cytotoxic effects of native peptides. As previous studies have reported, LL-37 (13–36), which includes an amphiphilic helix rich in positively-charged side chains, can effectively neutralise LPS and attenuate pathogen-induced inflammation [34,35]. In the present study, we designed a hybrid peptide (LL-37-TP5, LTP) by combing the active centre of LL-37 (13–36) with TP5. The hybrid peptide was hypothesized to have a stronger anti-inflammatory activity, longer half-life, and lower cytotoxicity than its parental peptides. Moreover, we also aimed to clarify the anti-inflammatory mechanisms of LTP by thoroughly exploring the molecular basis of its anti-inflammatory effects using a murine model of systemic LPS-induced inflammation.

## 2. Results

### 2.1. Peptide Design and Characterisation

As shown in Table 1, the hybrid peptide LTP was designed by combining the core functional region of LL-37 with TP5. All peptides were successfully synthesized by chemical methods. The molecular weight and amino acid sequence were confirmed to be correct. The designed hybrid peptide LTP had a higher net charge (+5) than TP5 (+1). However, compared with those of LL-37 (+6 and −0.724), LTP had a lower net charge and hydrophobicity. The mean hydrophobicity index of LTP (−0.821) was between those of LL-37 (−0.724) and TP5 (−1.8), which may indicate that C-L had a more balanced amphipathic characteristic.

### 2.2. Cytotoxicity on RAW264.7 Macrophage Cells

The cytotoxic activity of LTP on RAW264.7 macrophage cells was assessed by CCK-8 assay (Figure 1). Among the peptides, LTP exhibited a lower cytotoxicity than LL-37 and TP5. In addition, LTP exhibited no significant cytotoxicity towards RAW264.7 cells, even at the highest concentration of 60 μg/mL. These data indicated that LTP was suitable for the subsequent experiments.

### 2.3. Ex Vivo Stability of LTP in Plasma

The plasma concentration of each target peptide over time is shown in Figure 2. TP5 concentrations decreased by 95% after incubation for 5 min, consistent with previous research findings [36,37]. LTP showed a significantly prolonged degradation profile up to 720 min and exhibited a prolonged half-life (t1/2 of 240 min) compared with that of TP5 in plasma, whereas no statistical significance was found between LTP and LL-37 (Table 2).

### 2.4. Effect of LTP on Body Weight and Disease Activity Index

As expected, LPS-induced intestinal inflammation was availed by the disease activity index (DAI) [38,39] and clinically characterized by the presence of diarrhea, blood in the perianal region, and significant weight loss. Compared to mice in the LPS group pre-treated with LL-37, TP5 and LTP promoted a significant improvement in the DAI values (Figure 3). In addition, LTP was found to be more effective than LL-37 or TP5 in maintaining the DAI value.

### 2.5. The Protective Effects of LTP Against LPS-Induced Damage in Jejunum Tissue

Histological examination of jejunum tissues revealed that compared with the control group, the LPS group exhibited considerable tissue injury with extensive ulceration of the epithelium, oedema, structural damage to the mucous layer (Figure 4A), and a decreased villus height to crypt depth (V/C) ratio (Figure 4C). Overall, the LPS-induced intestinal damage was significantly attenuated by LTP pre-treatment. Chiu’s score in the LTP-pre-treated group was markedly lower than that in the LL-37-pre-treated group and TP5-pre-treated group (Figure 4B). Moreover, the V/C value in the LTP-pre-treated group was markedly higher than that in the TP5-pre-treated group, whereas no statistical significance was found between the LTP-pre-treated and LL-37-pre-treated groups (Figure 4C).

To characterise the inhibitory effect of LTP against inflammation in LPS-induced mice, the inflammatory markers TNF-α, IFN-γ, and IL-6 in mouse jejunum tissue were quantified by ELISA. Administration of LPS caused a significant elevation in the jejunum levels of the pro-inflammatory cytokines TNF-α, IL-6, and IFN-γ compared with those in the jejunum from the control group (Figure 5A–C). By contrast, all peptides attenuated the secretion levels of TNF-α, IFN-γ, and IL-6. Meanwhile, mice in the LTP supplemented group had lower TNF-α and IFN-γ concentrations than those in the LL-37 and TP5 supplemented groups (Figure 5A,B). However, there was no difference in IL-6 concentrations between the LTP-pre-treated group and the LL-37-pre-treated group (Figure 5C).

Immunohistochemistry results demonstrated that LPS triggered increased infiltration of CD177+ neutrophils into the jejunal lesion area compared with control (Figure 5D). Pretreatment with all three peptides reduced the infiltration of neutrophils compared with the group treated with LPS alone. LTP, the most active peptide, reduced this effect to the basal level. 

As depicted in Figure 5E, there was a significant increase in MPO (an indicator of jejunum infiltration by polymorphonuclear leukocytes) activity in jejunum tissue from LPS-treated mice compared with that in control mice, while the LTP-pre-treated group showed significantly reduced MPO activity compared with that in the LPS-treated alone group. In addition, MPO activity in the LTP-pre-treated group was markedly decreased compared with those in the LL-37-pre-treated and TP5-pre-treated groups. 

### 2.6. LTP Attenuated the LPS-Induced Disruption of Intestinal TJ Structure and Function

To investigate the effects of LTP on the functional integrity of mouse intestinal epithelium, TEER measurements were performed (Figure 6A). The TEER values in the LPS alone-treated group decreased significantly, indicating an increase in permeability and damaged intestinal barrier functioning. In contrast, pre-treatment with LTP resulted in a significant protective effect. Moreover, the TEER values in the LTP-pre-treated group were similar to those in the control group, which were significantly higher than those in the LL-37-pre-treated and TP5-pre-treated groups. These data demonstrate the notable role of LTP activation in minimising LPS-stimulated intestinal epithelial hyper-permeability.

To further clarify the protective mechanism of LTP on the LPS-induced disruption of TJs, the expression of TJ markers (ZO-1 and Occludin) were monitored by Western blotting (Figure 6B,C). The expression of TJ markers decreased in the mice treated with LPS alone compared with that in the controls. However, the administration of LTP markedly increased the expression of these TJ markers. To confirm these protective effects, ZO-1 in the jejunum tissues was detected by immunohistochemistry (Figure 6D). The expression of ZO-1 in jejunum tissue was significantly lower in the LPS-treated group than that in the control group. Notably, the addition of LL-37 and LTP, especially LTP, reduced these trends. In addition, TEM was used to detect the TJs between gut epithelial cells, and these results also supported the protective effect of LTP against LPS-induced damage in jejunum tissues (Figure 6E).

### 2.7. The Protective Effects of LTP Against LPS-Induced Oxidative Stress in Jejunum Tissue

Compared with the levels in the controls, LPS challenge led to high levels of MDA in the jejunum tissues, a trend which was reversed (*p* < 0.05) by LTP (Figure 7A). However, the activities of SOD (Figure 7B) and CAT (Figure 7C) decreased following LPS treatment in jejunum tissues, whereas the LTP-pre-treated group showed significantly increased SOD and CAT activities compared with those in the LPS-treated group. Compared with the parental peptides LL-37 and TP5, LTP had a marked ability to protect mouse jejunum against LPS-induced oxidative stress.

### 2.8. Effects of LTP on the NF-κB Signalling Pathway in LPS-Induced Mice

The immunohistochemistry results showed that LPS significantly increased the expression of TLR4 and phosphorylated AKT, IκB-α, and NF-κB (p65) compared with levels in the control group (Figure 8). Treatment with LTP effectively inhibited the phosphorylation of these proteins. Moreover, the expression of TLR4 and phosphorylated AKT, IκB-α, and NF-κB (p65) in the LTP pre-treated group was significantly lower than that in the groups pre-treated with the parental peptides. The suppressive effect of LTP appears to involve NF-κB translocation. These results suggest that the effect of LTP on the NF-κB signalling pathway plays a crucial role in the process by which LTP modulates the LPS-induced inflammation and barrier dysfunction in mice.

### 2.9. LPS Neutralization Activity of LTP in Vitro and in Vivo

In the LPS-treated group, the plasma LPS level sharply increased. However, pre-treatment of LPS-administered mice with LTP significantly reduced the plasma LPS level (Figure 9A). Moreover, the plasma LPS level in the LTP-pre-treated group was markedly lower than those in the groups treated with the parental peptides.

To determine whether LTP neutralized LPS, an LPS neutralization activity assay was performed in vitro. As shown in Figure 9B, LTP neutralized LPS in a dose-dependent manner. The LPS neutralization activity of LTP was about 1.5-fold less potent than that of polymyxin B (PMB), a cyclic hydrophobic peptide known to bind LPS [40], with 50% binding rate values of 1.5 μg/mL (LTP) and 1 μg/mL (PMB). In addition, the LPS neutralization activity of LTP was stronger than those of its parental peptides LL-37 and TP5.

## 3. Discussion

LPS, a major component of the cell wall of Gram-negative bacteria [41], can induce severe intestinal inflammation and cause local and systemic complications, such as abdominal pain, diarrhoea, rectal bleeding, weight loss, malnutrition, and fever [3]. Corticosteroids used in the clinical treatment of intestinal inflammation are typically based on their anti-inflammatory activity [11,12,13]. However, their side effects, such as central adiposity, insulin resistance, and gastrointestinal complications, have hampered their clinical development [15,16]. Thus, there is an urgent need to identify new medicines to treat this severe disease. 

Recently, native anti-inflammatory peptides, such as LL-37 and TP5, have shown enormous potential in the treatment of a range of inflammatory diseases [27,28,42]. However, several obstacles, including potential cytotoxicity [29], poor anti-inflammatory activity based on peptide concentration, and weak physiological stability [43], reduce their clinical potential. 

Hybridising is an effective strategy for designing novel functional peptides because the hybrid can possess the best elements of the various parental peptides [44,45]. However, to our knowledge, scientific research is lacking on the rational design of peptide structures and the correlations between this type of information and anti-inflammatory activities. Based on previous findings, positive charge and hydrophobicity are required for a peptide to have anti-inflammatory activity, as these characteristics affect the LPS neutralization activity and cellular uptake ability of the peptides [46,47]. In the present study, we designed a hybrid peptide (LTP) by combining the active center of LL-37 (13–36) [34,35] with TP5. The primary structure characteristic results showed that the mean hydrophobicity index and the positive charge of LTP (−0.821, +5) were between those of LL-37 (−0.724, +0.6) and TP5 (−1.8, +1), which indicated that LTP had a more balanced positive charge and hydrophobicity than its parental peptides. These results suggested that LTP may have a stronger potential anti-inflammatory activity than its parental peptides based on previous studies [46,47]. 

Cytotoxicity is often thought to be a barrier for the therapeutic use of peptides. Thus, it was important to evaluate toxicity. In this study, the proliferation assays showed that LTP had a lower cytotoxicity than its parental peptides. This decreased cytotoxic activity may be due to the rational hydrophobicity of the hybrid peptide, which is similar to that described in other studies [45]. 

Considering the short half-life of TP5 in plasma, repeated injections and a long treatment duration are necessary to improve clinical efficacy [43]. Thus, it was important to prolong the half-life of the peptide. In the present study, the results indicated that the in-vitro half-life of LTP was prolonged to 240 min in rat plasma.

The present study showed that the symptoms of intestinal inflammation, such as weight loss, neutrophil infiltration, histological features of multiple erosions, and inflammatory intestinal mucosal changes, induced by LPS in mice are similar to those of human IBD [2,48,49]. However, LTP effectively attenuated weight loss, diarrhea, and histological damage, including ulceration of the epithelium and decreased villus height to crypt depth ratio, caused by LPS. Infiltration of activated neutrophils, one of the most representative histological features of intestinal inflammation [50,51], was significantly increased in LPS-treated mice; however, pre-treatment of mice with LTP prevented the infiltration of activated neutrophils caused by LPS. MPO activity is an index of neutrophil infiltration and inflammation because it is directly proportional to the neutrophil concentration in the inflamed tissue [52]. Consistently, treatment with LPS markedly increased MPO activity in the jejunum. In contrast, MPO activity was markedly reduced in LTP-pre-treated mice, and it was significantly lower than that in the Tα1 and LL-37 pre-treated groups. 

Intestinal inflammation is characterised by the excessive release of inflammatory cytokines. Inflammatory cytokines, such as TNF-α, can amplify the inflammatory cascade by triggering the accumulation and activation of leukocytes [53]. Our data showed that pre-treatment with all peptides attenuated the secretion levels of TNF-α, IFN-γ, and IL-6. Meanwhile, the LTP supplemented group had lower TNF-α and IFN-γ concentrations than the LL-37 or TP5 supplemented group.

The gut epithelial barrier is a physical and metabolic barrier against toxins and pathogens between the lumen and the circulatory system [54]. We evaluated the effects of LTP on gut epithelial barrier function in vivo. TEER of the epithelium is an indicator of intestinal epithelial integrity and permeability [55]. In this study, LTP effectively attenuated the LPS-induced decrease in the TEER value. In addition, the TEER values of the LTP-pre-treated group were significantly higher than those in the LL-37- and TP5-pre-treated groups. Intercellular TJ proteins, important components of the intestinal epithelial barrier [56], are responsible for limiting the paracellular movement of compounds across the intestinal mucosa [57]. Loss of TJ proteins leads to increased intestinal permeability and impairment, eventually contributing to gut inflammation diseases [58]. Our data indicated that LTP effectively alleviated the LPS-induced decrease in the expression of the TJ proteins ZO-1 and occludin, and this effect was more potent than that by LL-37 and TP5. In addition, to further analyse the TJs between gut epithelial cells, TEM was performed, and the results supported the protective effect of LTP against LPS-induced damage in jejunum tissues.

Collectively, these results indicate that LTP had a higher anti-inflammatory potency and stability than its parental peptides, while also having minimal cytotoxicity. To identify the mechanisms of the observed anti-inflammatory effects in LPS-treated mice, a comprehensive and detailed analysis was conducted.

LPS, a major constituent of the outer membrane of Gram-negative bacilli, can upregulate approximately 100 different genes, such as those for proinflammatory cytokines, signalling molecules, and transcriptional regulators [59]. Therefore, LPS has strong inflammatory activity and plays an important role in the pathogenesis of intestinal inflammation [2]. Our data showed that the LTP-pre-treated mice had significantly lower plasma LPS levels than the LPS-only treated mice. Meanwhile, the LTP supplemented group had lower LPS levels than the LL-37 or TP5 supplemented group. To determine whether LTP neutralized LPS, an LPS neutralization activity assay was performed in vitro. The results showed that LTP can almost completely neutralise LPS at a concentration of 8 μg/mL. The LPS neutralization activity of LTP was similar to that of PMB, a cyclic hydrophobic peptide known to bind LPS [48], and was markedly more potent than its parental peptides. Thus, the present study indicated that LTP can significantly attenuate intestinal inflammatory effects by binding LPS and then preventing its intracellular accumulation or translocation across the cell membrane. 

LPS is responsible for the synthesis of reactive oxygen species (ROS), and the overproduction of ROS may cause considerable oxidative stress [8]. LPS also can lead to an increase in MDA and a reduction in endogenous antioxidant defences [60]. Meanwhile, previous studies have shown that oxidative stress plays an important role in the pathogenesis, progression, and severity of intestinal inflammation [10]. The present study showed that pre-treatment with LTP significantly increased SOD and CAT activities and markedly decreased the MDA leaves. In addition, compared with the parental peptides LL-37 and TP5, LTP had a marked ability to protect the mouse jejunum against LPS-induced oxidative stress. The results of this study suggested that LTP can attenuate LPS-induced intestinal inflammation by protecting the mouse jejunum against oxidative stress.

NF-κB, a key transcriptional regulator of the inflammatory response, can be activated by different extracellular stimuli, such as oxidative stress, LPS, and cytokines [61]. Thus, NF-κB plays a crucial role in the development of the intestinal inflammatory process [62]. In the present study, the expression of the major proteins involved in the NF-κB pathway were assessed to clarify the anti-inflammatory mechanism of LTP in intestinal inflammation. Our data showed that LTP effectively inhibited the activation of NF-κB signalling by suppressing the expression of the TLR4 and the phosphorylation of AKT, IκB-α, and NF-κB.

## 4. Materials and Methods 

### 4.1. Hybrid Peptide Design

The hybrid peptide LL-37-TP5 (LTP) was constructed by combining the active center of LL-37 (13–36) with TP5. The amino acid sequences of the parental and hybrid peptides are listed in Table 1.

### 4.2. Sequence Analysis of the Hybrid Peptide

The mean hydrophobicity and net charge of the peptides were calculated online via ProParam (ExPASy Proteomics Server: http://www.expasy.org/tools/protparam.html). 

### 4.3. Peptide Synthesis

The peptides used in this study, LL-37, TP5, and LTP, were synthesized and purified by KangLong Biochemistry (Jiangsu, China). The purity of the peptides was determined by HPLC and mass spectrometry, revealing that more than 95% purity was achieved. The peptides were dissolved in endotoxin-free water and then stored at −80 °C.

### 4.4. Cell Culture

The murine macrophage cell line RAW264.7 was purchased from the Shanghai Cell Bank, the Institute of Cell Biology, China Academy of Sciences (Shanghai, China). RAW264.7 cells were cultured in Dulbecco’s modified Eagle’s medium (DMEM; Hyclone, Logan, UT, USA) supplemented with 10% (*v*/*v*) foetal bovine serum (Bioscience) and 1% (*v*/*v*) penicillin/streptomycin (Hyclone) in a 37 °C humidified incubator containing 5% CO_2_.

### 4.5. Cell Viability Assay

The viability of peptide-treated RAW264.7 cells was measured using the Cell Counting Kit-8 (CCK-8) Assay Kit (Dojindo, Japan) [63]. The cells were pre-seeded on a 96-well plate at a density of 3 × 10^4^ cells/mL overnight. The cells were either treated with various concentrations of peptides (LL-37, TP5, and LTP) or incubated without peptides at 37 °C and 5% CO_2_ for 24 h or 72 h. Each well was then incubated with 10 μL CCK-8 solution for 4 h in the dark. The absorbance at 450 nm was measured with a microplate reader. Cell viability was calculated as:
Cell viability (%) = (OD_450 (sample)_/OD450_(control)_) × 100%(1)where OD_450 (sample)_ is the absorbance at 450 nm of the cells with peptides treated, and OD_450 (control)_ is the absorbance at 450 nm of the cells without peptides treated.

### 4.6. Ex Vivo Stability of LTP in Plasma

Rat plasma was collected from healthy adult rats by centrifugation of whole blood samples. LL-37, TP5, and LTP were directly added to rat plasma (10 μg/mL), after which the plasma samples were incubated in a 37 °C water bath. At different times, the samples were collected into pre-chilled tubes which contained 1 mL of acidic acetone (hydrochloric acid/acetone/H_2_O, 1:40:5, by volume). Subsequently, the mixture was centrifuged at 2 × 10^4^ rpm at 4 °C for 20 min. The precipitates were dried in vacuo, after which the dried precipitates were dissolved in 0.5 mL of 1 M acetic acid. The peptide analysis was carried out according to the protocol for the study of TP5 using HPLC [64]. The half-lives of the peptides were calculated by a logarithm-linear regression analysis of the peptide concentrations.

### 4.7. Animal Model

Sixty C57/BL6 male mice (6–8 weeks of age) were purchased from Charles River (Beijing, China). All animals were maintained in a specific pathogen-free (SPF) environment at 22 ± 1 °C with relative 55 ± 10% humidity, and the mice had free access to food and drinking water during the whole period (1 week). All of the animal experiments were approved by the Institutional Animal Care and Use Committee of China Agricultural University and were performed in accordance with guidelines set forth by the Care and Use of Laboratory Animals of the Ministry of Science and Technology of China (certificate of the Beijing Laboratory Animal employee, ID: 18086).

The mice were randomly distributed into five groups of 12 each: control, LPS (*E. coli*, O111:B4, Sigma-Aldrich, USA) treatment, LL-37 pretreatment followed by LPS treatment (LL-37 + LPS), TP5 pretreatment followed by LPS treatment (TP5 + LPS), and LTP pretreatment followed by LPS treatment (LTP + LPS). The different peptides (10 mg/kg mouse weight) were injected intraperitoneally once daily for 7 d, whereas an equal volume of sterile saline was injected to the control and LPS-treated groups. On day 7, mice in the LPS, LL-37 + LPS, TP5 + LPS and LTP + LPS groups were intraperitoneally injected with LPS (10 mg/kg mouse weight) 1 h after the saline or the peptide treatment, and the control group was injected with an equal volume of saline. The mice were euthanised by cervical dislocation 6 h after the intraperitoneal injection of LPS or saline, and samples of the intestine were collected for analysis.

### 4.8. Evaluation of Intestinal Inflammation 

LPS-induced intestinal inflammation was availed by DAI [38,39]. The DAI was assessed according to a standard scoring system [65]. Body weight, stool consistency, and occult blood in the stool were recorded. For body weight, a score of 0 was assigned for no weight loss, 1 for 1–5% weight loss, 1 for 5–10% weight loss, 4 for 10–20% weight loss, and 4 for more than 20% body loss. Stool consistency was scored as: 0, well-formed pellets; 2, pasty and semi formed stools that did not adhere to the anus; and 4, liquid stools that adhered to the anus. For occult blood, a score of 0 was assigned for no blood, 2 for occult blood, and 4 for gross bleeding. These scores were added together and then divided by three, which resulted in DAIs ranging from 0 (healthy) to 4 (maximal intestinal inflammation). 

### 4.9. Histopathology and Immunohistochemistry

Intestinal jejunum tissues were fixed in 4% paraformaldehyde and embedded in paraffin. After embedding, 5-μm-thick sections were stained with haematoxylin and eosin (H&E), and the images were acquired with a microscope (Leica, Wetzlar, Germany). Villous height and crypt depth were measured with Image-Pro software (MediaCybernetics, MD, USA) [66]. LPS-induced intestinal injury was evaluated using Chiu’s score [67] according to changes of the villus and glands of the jejunal mucosa.

For immunohistochemical analysis of CD177^+^, nonspecific binding sites were blocked with PBS containing 1% *w*/*v* BSA for 1 h. Subsequently, anti-CD177^+^ antibodies (Santa, USA) were added at a dilution of 1:100 and incubated overnight at 4 °C. Samples were treated with horse-radish peroxidase (HRP)-conjugated rabbit anti-goat IgG (JIR, USA) at a ratio of 1:100 after washing with PBS. Samples were left to incubate at 4 °C for 1 h and washed with PBS. Then, 3, 3′-diaminobenzidine (DAB; DAKO, USA) was added, and the slices were stained with Harris haematoxylin. Finally, the samples were dehydrated in a gradient alcohol series (70–100%) and cleared in xylene. The slides were mounted in neutral balsam.

The expression levels of the intercellular tight junction protein zonula occludens-1 (ZO-1) were evaluated. Jejunum sections were blocked with PBS containing 1% *w*/*v* BSA for 30 min, after which the sections were incubated with anti-ZO-1 (Abcam, USA) overnight at 4 °C. Slices were washed with PBS and then incubated with a tetramethylrhodamine isothiocyanate-conjugated secondary Ab for 1 h at room temperature in the dark. The samples were then mounted in medium containing the nuclei marker 4,6-diamidino-2-phenylindole (DAPI) (Servicebio, Wuhan, China). Images were taken immediately under a fluorescence microscope.

### 4.10. ELISA

Levels of tumour necrosis factor-alpha (TNF-α), interferon-gamma (IFN-γ), and interleukin-6 (IL-6) in the jejunum were detected using ELISA kits (eBioscience, San Diego, CA, USA) according to the manufacturer’s instructions.

The activity of myeloperoxidase (MPO) in the jejunum was detected with ELISA kits (Boster Wuhan, China). The samples were measured according to the manufacturer’s instructions.

### 4.11. Antioxidant Capacity and Antioxidant Enzymes

The concentration of malondialdehyde (MDA) in the jejunum was detected using kits (Cell Biolabs Inc., San Diego, CA, USA). The activities of superoxide dismutase (SOD) and catalase (CAT) in the jejunum were determined using diagnostic kits (Oxis International Inc., Portland, OR, USA). The samples were measured according to the manufacturers’ instructions.

### 4.12. Western Blotting

Whole protein of jejunal tissues was obtained with a whole protein extraction kit (KeyGEN Biotech, Nanjing, China). The protein concentration was measured with a BCA kit (KeyGEN Biotech, Nanjing, China). Protein samples were separated by 10% SDS-PAGE and transferred onto PVDF membranes (Bio-Rad). The membrane was blocked with 5% non-fat dried milk in 0.05% TBST and then immunoblotted overnight at 4 °C with primary specific antibodies against Toll-like receptor 4 (TLR4), NF-κB (p65), p-NF-κB (p-p65), AKT, p-AKT, IκB-α, p-IκB-α, ZO-1, Occludin, and β-actin (Santa Cruz, CA, USA). After washing with TBST, the membranes were incubated with HRP-conjugated secondary antibody (HuaAn, Hangzhou China). The proteins were visualised with a ChemiDoc MP Imaging System (Bio-Rad, Hercules, CA, USA).

### 4.13. Transmission Electron Microscopy (TEM)

The tight junctions (TJs) between gut epithelial cells were observed by TEM. Jejunum specimens were excised with a scalpel and fixed in 2.5% glutaraldehyde for 4 h at 4 °C. Subsequently, the specimens were treated with osmic acid and embedded in Epon. Ultrathin sections were acquired and stained with uranyl acetate and alkaline lead citrate. Images were taken under a transmission electron microscope (Model H-7650, HITACHI, Tokyo, Japan).

### 4.14. Electrophysiology Measurements

The transepithelial electrical resistance (TEER) values of jejunal membranes were determined using an in-vitro diffusion chamber method using stripped mouse jejunal membranes. The jejunal segments were mounted in a diffusion chamber with an exposed surface area of 1.78 cm^2^ after removing the underlying muscularis of the jejunal membranes. Ussing chambers were equipped with two pairs of electrodes connected to the chambers by 3 M KCl/3.5% agar bridges for measuring the potential difference (PD) and the short-circuit current (Isc), respectively, and then, total electrical resistance (RT) was calculated using Ohm’s law: RT = PD/ Isc [68].

### 4.15. Neutralization of LPS

The neutralization of LPS by the peptides was detected using a quantitative Chromogenic End-point Tachypleus Amebocyte Lysate (CE TAL) assay with the QCL-1000 kit (Xiamen, China). Experiments were carried out according to the institutional guidelines. A constant concentration of LPS (1.0 U/mL final concentration) was incubated with different concentrations of polymyxin B (PMB) or the peptides (0 to 64 μg/mL final concentration) at 37 °C for 15 min in the wells of pyrogenic sterile microliters plates. Subsequently, the mixtures were incubated at 37 °C for 6 min in the presence of TAL assay reagent, after which the absorbances were measured at 540 nm.

### 4.16. Statistics

All the data are expressed as mean values ± standard error from at least three independent experiments. Experimental data were carried out by one-way ANOVA and post-hoc analysis by Duncan’s test with SPSS21.0. Significance was claimed with *p* values ≤ 0.05. NS: *p* > 0.05, *: *p* ≤ 0.05, **: *p* ≤ 0.01, ***: *p* ≤ 0.001, ****: *p* ≤ 0.0001.

## 5. Conclusions

In this study, we proposed a feasible approach for the design of new anti-inflammatory peptides via the hybridization of different native peptides with anti-inflammatory activity. This method enhances the native peptides’ anti-inflammatory potency and stability, while simultaneously minimising cytotoxicity. These advantageous properties might be the result of the successful creation of a proper hydrophobic-cationic balance. The hybrid peptide LTP effectively inhibited LPS-induced impairment of jejunum epithelium tissues and infiltration of leukocytes. Additionally, LTP decreased the levels of TNF-α, IFN-γ, and IL-6, increased the expression of ZO-1 and occludin, and reduced permeability in the jejunum. Our study also confirmed that the anti-inflammatory effects of LTP on LPS-induced intestinal inflammation may be associated with the neutralization of LPS, the inhibition of oxidative stress, and the inhibition of the NF-κB signalling pathway. Our findings provide a strong rationale for the further development of active anti-inflammatory agents for treating intestinal inflammation.

## Figures and Tables

**Figure 1 ijms-20-03974-f001:**
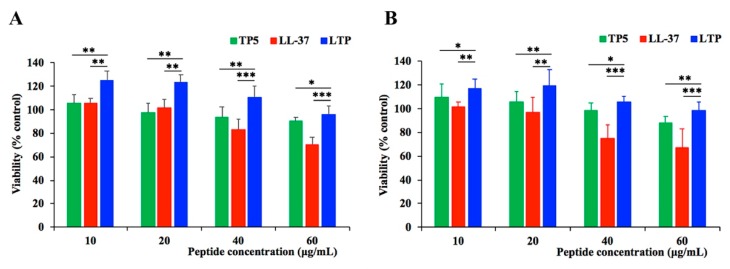
Cell proliferation rates of RAW264.7 cells in the absence or presence of hybrid peptide and parental peptides. RAW264.7 cells were pre-seeded in DMEM medium overnight. The cells were treated with various concentrations of peptides or without peptides at 37 °C and 5% CO_2_ for 24 h (**A**) or 72 h (**B**). They were incubated with CCK-8 solution for 4 h, and then, they were measured at 450 nm. Data are given as the mean value ± standard error from eight biological replicates. *: *p* ≤ 0.05, **: *p* ≤ 0.01, ***: *p* ≤ 0.001.

**Figure 2 ijms-20-03974-f002:**
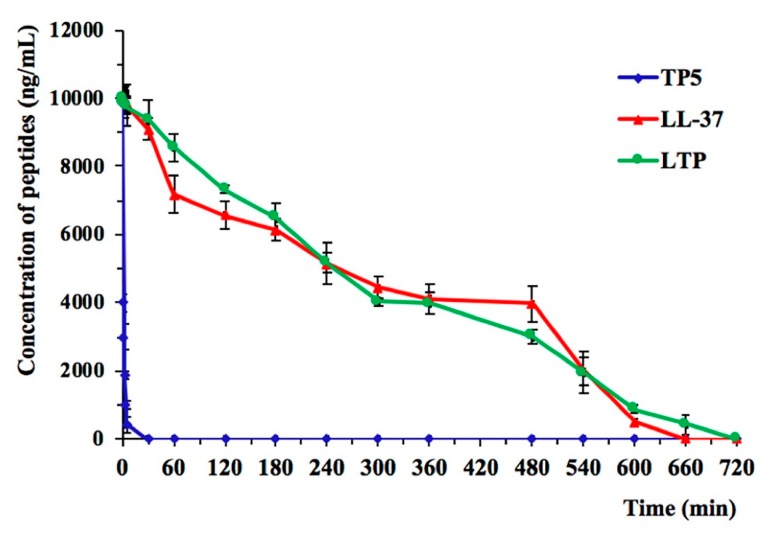
The mean plasma concentrations of LL-37, TP5, and LTP over times. The concentration of each target peptide in plasma in vitro at the selected times was quantified by HPLC. Data are given as the mean value ± standard error from three biological replicates.

**Figure 3 ijms-20-03974-f003:**
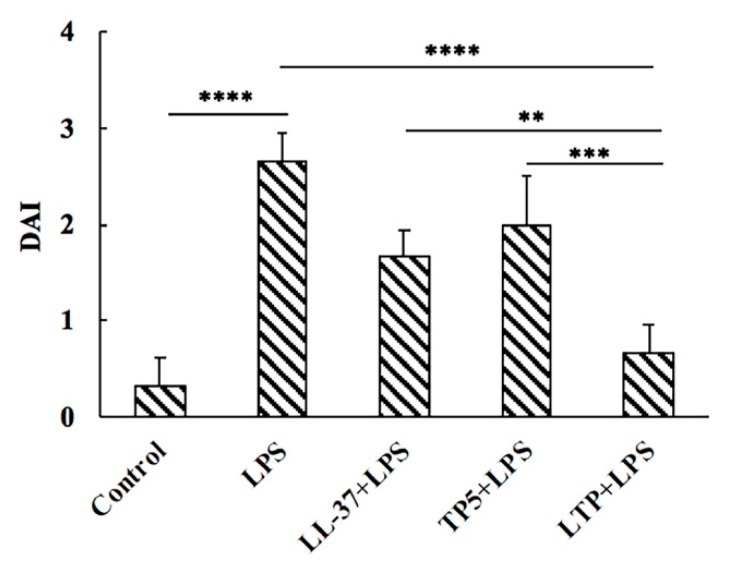
LTP improved disease activity index (DAI). Different peptides (10 mg/kg) were injected into the mice once daily for 6 days, whereas the control and LPS-treated groups were injected with an equal volume of sterile saline. On day 6, mice in the LPS and peptide-pretreated groups were injected with LPS (10 mg/kg) 1 h after the peptide or saline treatment. The control group was injected with an equal volume of saline. The body weights and DAI of the mice were recorded before and after the experiment. Data are given as the mean value ± standard error from 12 biological replicates. **: *p* ≤ 0.01, ***: *p* ≤ 0.001, ****: *p* ≤ 0.0001.

**Figure 4 ijms-20-03974-f004:**
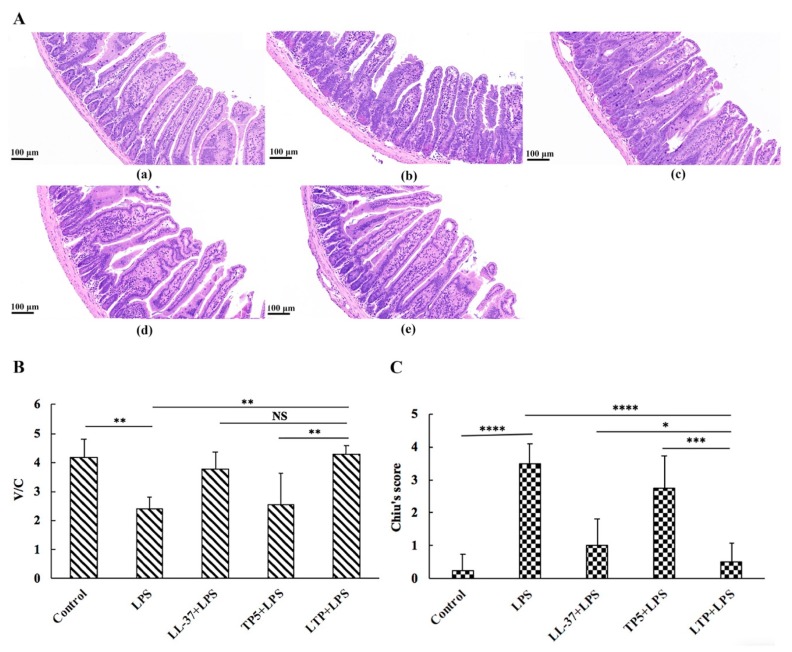
The protective effects of LTP against LPS-induced clinical symptoms in the jejunum. Representative H&E-stained sections from the (**A**-**a**) control, (**A**-**b**) LPS, (**A**-**c**) LL-37 + LPS, (**A**-**d**) TP5 + LPS, and (**A**-**e**) LTP + LPS groups. Bar, 100 μm. (**B**) The effect of LTP on Chiu’s scores. Chiu’s score is comprised of changes of the villus and glands of the jejunal mucosa. (**C**) Protective effect of LTP on the ratio of villus height to crypt depth (V/C). Data are presented as the mean value ± standard error from 12 biological replicates. NS: *p* > 0.05, *: *p* ≤ 0.05, **: *p* ≤ 0.01, ***: *p* ≤ 0.001, ****: *p* ≤ 0.0001.

**Figure 5 ijms-20-03974-f005:**
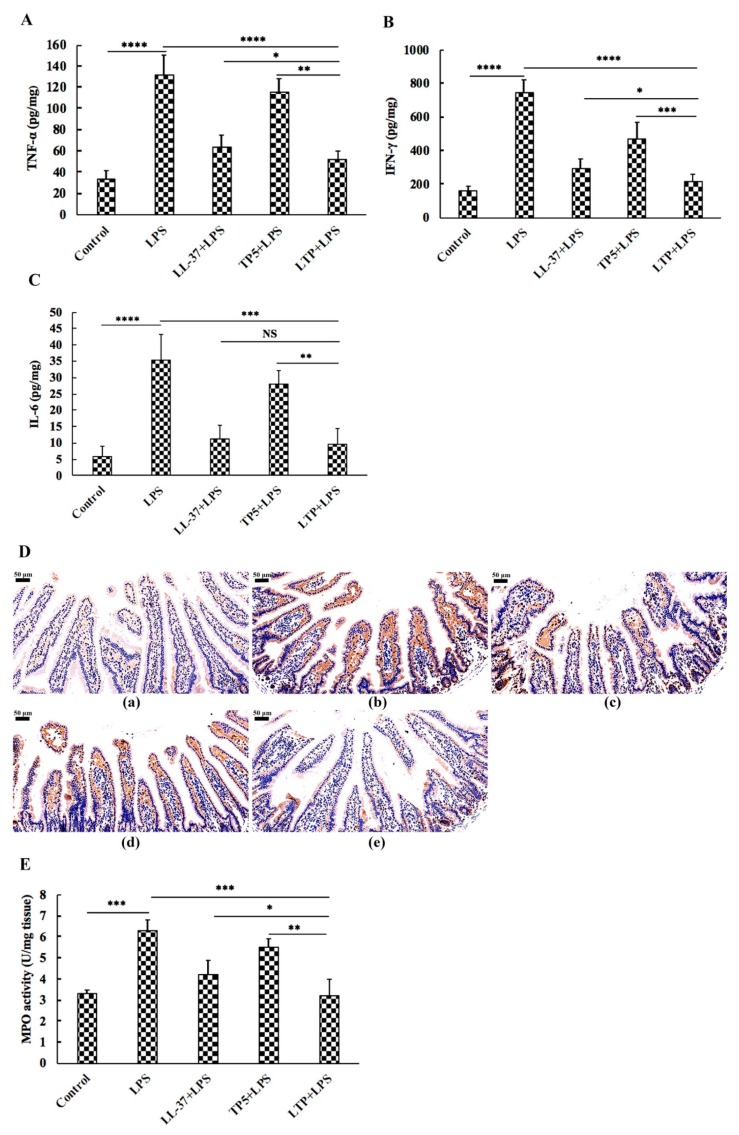
The protective effects of LTP on the inflammatory response. After treatment with 10 mg/kg LPS in the presence or absence of peptides, protein levels of TNF-α (**A**), IFN-γ (**B**), and IL-6 (**C**) in mouse jejunum tissue were quantified by ELISA. (**D**) Representative images of CD177+. Bar, 50 μm. (**D**-**a**) control, (**D**-**b**) LPS, (**D**-**c**) LL-37 + LPS, (**D**-**d**) TP5 + LPS, and (**D**-**e**) LTP + LPS. (**E**) Enzymatic activities of MPO. Data are presented as the mean value ± standard error from 12 biological replicates. NS: *p* > 0.05, *: *p* ≤ 0.05, **: *p* ≤ 0.01, ***: *p* ≤ 0.001, ****: *p* ≤ 0.0001.

**Figure 6 ijms-20-03974-f006:**
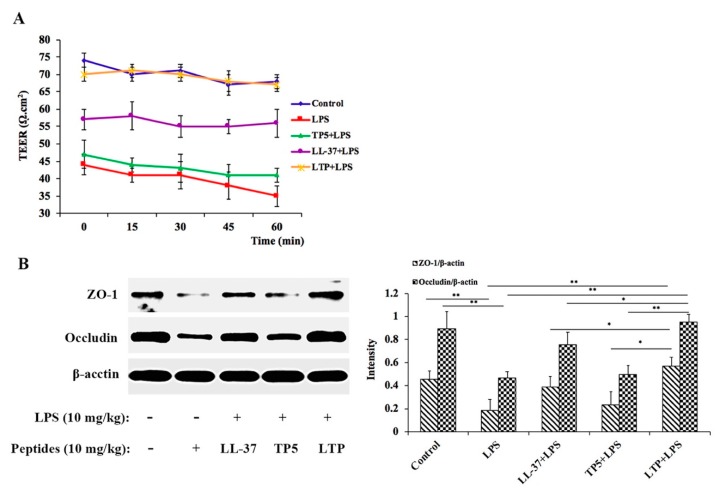
The protective effects of LTP on the intestinal barrier. (**A**) TEER values of mouse jejunum epithelium were determined ex vivo in Ussing chambers. (**B**) Expression of TJ markers (ZO-1 and Occludin) were monitored by Western blotting. (**C**) Jejunum sections were stained for ZO-1 (red) and nuclei (blue) and analysed by immunofluorescence microscopy. Bar, 50 μm. (**C**-**a**) Control, (**C**-**b**) LPS, (**C**-**c**) LL-37 + LPS, (**C**-**d**) TP5 + LPS, and (**C**-**e**) LTP + LPS. (**D**) The TJ structure of jejunal epithelium was confirmed by TEM. Bar, 2 μm. (**D**-**a**) control, (**D**-**b**) LPS, (**D**-**c**) LL-37 + LPS, (**D**-**d**) TP5 + LPS, and (**D**-**e**) LTP + LPS; the wider intervals and more blurred desmosomes (black arrowheads) between the intestinal epithelial cells are indicated. Data are given as the mean value ± standard error from at least three biological replicates. *: *p* ≤ 0.05, **: *p* ≤ 0.01.

**Figure 7 ijms-20-03974-f007:**
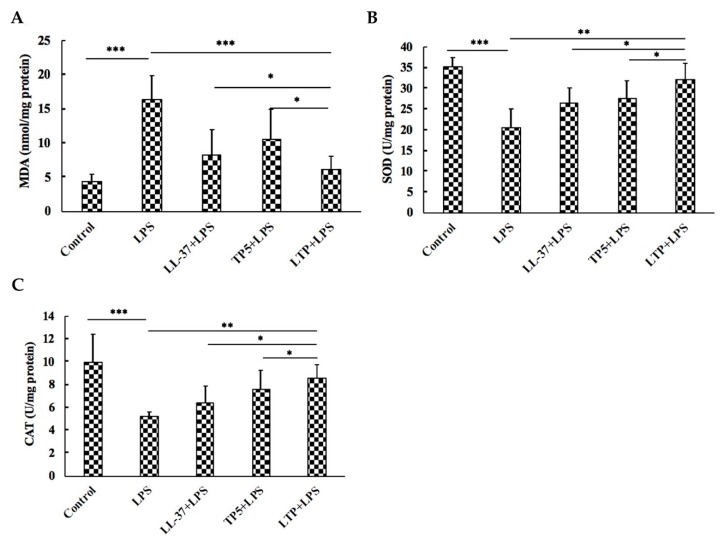
Effects of LTP on oxidative stress markers in the mouse jejunum. (**A**) MDA; (**B**) SOD activity; (**C**) CAT activity. Data are presented as means ± standard error from 12 biological replicates. *: *p* ≤ 0.05, **: *p* ≤ 0.01, ***: *p* ≤ 0.001.

**Figure 8 ijms-20-03974-f008:**
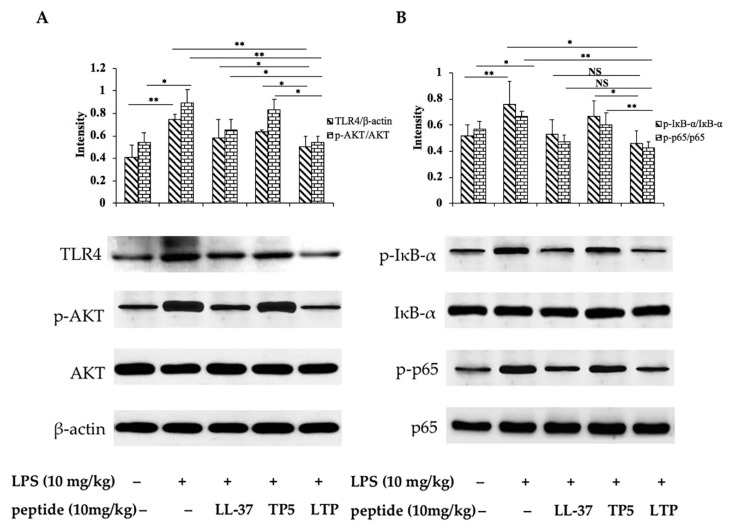
Inhibitory effect of LTP on the NF-κB signalling pathways in mice. Phosphorylated and total protein levels of TLR4, AKT, and β-actin (**A**); IκB-α and p65 (**B**) in jejunal tissues were measured by Western blot analyses. Data are presented as the mean value ± standard error from three biological replicates. NS: *p* > 0.05, *: *p* ≤ 0.05, **: *p* ≤ 0.01.

**Figure 9 ijms-20-03974-f009:**
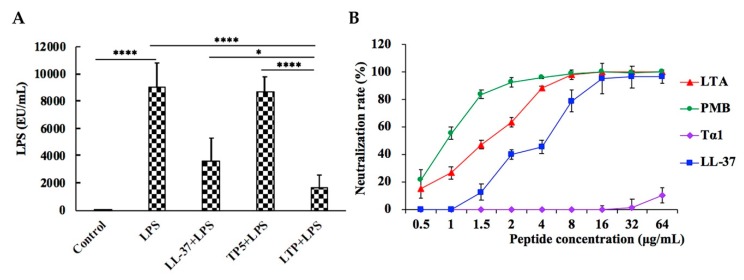
LPS neutralization activity of LTP in vitro and in vivo. (**A**) The LPS concentration in mouse plasma. (**B**) In-vitro LPS neutralization by LTP. LPS neutralization activity of LTP, LL-37, TP5, and PMB were determined using the chromogenic in vitro TAL assay. Data are presented as the mean value ± standard error from three biological replicates. *: *p* ≤ 0.05, ****: *p* ≤ 0.0001.

**Table 1 ijms-20-03974-t001:** Key physicochemical parameters of the parental and hybrid peptides.

Peptides	Sequence	H ^1^	Net Charge
LL-37	LLGDFFRKSKEKIGKEFKRIVQRIKDFLRNLVPRTES	−0.724	+6
TP5	RKDVY	−1.8	+1
LTP	IGKEFKRIVQRIKDFLRNLVPRTERKDVY	−0.821	+5

^1^ The mean hydrophobicity (H) is the total hydrophobicity (sum of all residue hydrophobicity indices) divided by the number of residues.

**Table 2 ijms-20-03974-t002:** Half-life of LTP in plasma.

Peptide	LL-37	TP5	LTP
**t_1/2_ (min)**	242.47 ± 45.09 ^a^	1.32 ± 0.24 ^b^	240.03 ± 55.14 ^a^

Data are given as the mean value ± standard error from three biological replicates. ^a, b^ Means with different superscripts within the same row differ (*p* < 0.01).

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
