# Peer review of "A Novel Peptide Ameliorates LPS-Induced Intestinal Inflammation and Mucosal Barrier Damage via Its Antioxidant and Antiendotoxin Effects"

_ijms, 2019, doi:10.3390/ijms20163974_

Round 1

Reviewer 1 Report

The author's have provided new data and significantly improved the manuscript addressing previous questions adequately.

However please address below suggestions

Explain the in detail scoring index for DAI. If weight loss was recorded in the DAI , the weight change graph becomes redundant and should be removed.

Authors need to carefully improve the quality of the graphs (eg same size) to make the it look more clear and easy to read.

Author Response

1.Thank you for the review. We have added the relative descriptions of disease activity index (DAI) as you suggested, and we have added the relative descriptions to lines 405-413 of the revised manuscript. Mean while, we have removed the weight change grough as you suggested.

2.Thanks for your comments. The figure quality has been improved accordingly

Reviewer 2 Report

The authors answered to the majority of questions asked from the two reviewers. I therefore consider the manuscript suitable for publication without the request of additional experiments, however if suggest a minor revision to improve the quality of several graphs before the publication.

The legend of Y-axis in several graphs are not in the middle, I suggest to uniform all graphics of Figure 1A-B, Figure 3B, Figure 4 B-C, Figure 5 A-B, Figure 7 A-B-C and Figure 8 A-B and Figure 9 A-B.

Author Response

Thank you for the review. We cordially appreciate your suggestions and appreciation.All the figure quality has been improved accordingly.

This manuscript is a resubmission of an earlier submission. The following is a list of the peer review reports and author responses from that submission.

Round 1

Reviewer 1 Report

1.Doubtful of the method used to induce in intestinal inflammation in the in vivo experiments.

 There are no strong experimental models to show systemic LPS injection leads to intestinal inflammation. A true intestinal inflammation is by chemical methods (DSS, TNBS) or spontaneous colitis models.  Systemic LPS might lead to temporary barrier impairment but not to the extent of a disease state.

A disease activity index (Stool consistency and blood in stool) has not been done to show the extent of intestinal inflammation.  Further doubts on if the mice developed intestinal inflammation.

2. Figure 1 & 2- The text mentions there significance, however there are no significance on the graph and legend. Please indicate significance on Graph.

3. Colon weight change – very high S.Ds, Please redo significance analysis.

4. For all figures the significance shown is misleading. Please compare all bars to the control. Please covert all SDs to the standard S.E.

5. The H& E images are distorted and not representative of the text description especially for LPS group. Please provide more suitable images.

Better representation of intestinal inflammation is seen at the colonic tissue, not clear why the jejunum sections were chosen. Please include histopathology of the colon as well.

6. All images are over corrected and cellulary structures are distorted. Please modify these images to more normal settings.

7. Authors claim LTP effectively inhibited the activation of NF-κB signalling by suppressing the phosphorylation of AKT, IκB-α, and NF-κB. However, this is a very generic pathway and no functional studies are done to show which receptors are activated by LTP. 

8. LPS neutralization activity of LTP – In vivo testing was done as pre-treatment and invitro testing was done as cotreatment. A cotreatment method aligns with the pathway shown. Then why was in vivo done a pre-treatment?

Reviewer 2 Report

The manuscript demonstrates that a novel peptide ameliorates LPS-induced intestinal inflammation via its anti-oxidant and antiendotoxin effects. The article described an improvement of native peptides by reducing their cytotoxicity and their poor anti-inflammatory activity. For this reason the novelty is mild, but with an important innovation thanks to the hybridization.

However, prior to full acceptance the following comments should be addressed or considered.  

1) The authors say that cytotoxicity is often a barrier for therapeutic use of peptides, suggesting that the valuation of cytotoxicity is an important issue to study.

 a)    Of note, the authors clearly demonstrated that LTP had a lower cytotoxicity compared to LL-37 and TP5. However, the authors have done the proliferation assay only at 24h, as written in the materials and method section. I first suggest to repeat the experiments at different time points (48h and 71h) to be sure that LTP is devoid of any effect on viability and to be sure that the cytotoxic effect is not delayed.

       b) The previous reports on LL-37 cytotoxicity has been done in human osteoblast-like      cell line (MG63, Anders et al., 2018). I suggest the authors to compare the proliferation assay in these cells using LL-37, TP5 and LTP.

 2) The novel peptide LTP has similar PD effect (e.g. intestinal length, body weight, etc) compare to LL-37, the authors must discuss the improvement of LTP in more detail.

 3) In Figure 8, the authors demonstrated that LTP has an anti-inflammatory effect by reducing NF-kB-signalling pathway. It seems unclear while they not found a reduction in total IkBa, since they observed an increase of pP65. The authors might discuss this in the text.

4) General comment:  the graphs and figures are not of good quality.  For example, in Figure 3, the text of Y-axis is not at the middle, the words overlap the error bars. I kindly suggest the authors to be careful of details.

 5) Minor comment: in Figure 1, no statistics are present.